# Features and predictive value of 6-min walk test outcomes in interstitial lung disease: an observation study using wearable monitors

Jiaying Li,[1] Xiaoyan Li,[2] Miaozhen Deng,[2] Xinyin Liang,[2] Huiqun Wei,[2] Xiaobing Wu  [3]

[1]School of Nursing, Li Ka Shing Faculty of Medicine, The University of Hong Kong, Hong Kong, Hong Kong
[2]Guangzhou Institute of Respiratory Health, First Affiliated Hospital of Guangzhou Medical College, Guangzhou, Guangdong, China
[3]Department of Internal Medicine, First Affiliated Hospital of Guangzhou Medical College, Guangzhou, Guangdong, China

**Correspondence to**
Dr Xiaobing Wu;
wuxiaobing_gz@163.com

## ABSTRACT

**Objectives** To describe 6-min walk test (6MWT) outcomes, and to investigate their correlations with cardiopulmonary and lung function among patients with interstitial lung disease (ILD) which was not limited to idiopathic pulmonary fibrosis.

**Methods** We collected patients' demographic data and obtained minute-by-minute 6MWT outcomes. Modified Borg scale was employed to assess patients' dyspnoea, whereas New York Heart Association (NYHA) classification and pulmonary function test were used to evaluate patients' cardiopulmonary functions.

**Results** Heart rate (HR) exhibited a continuous upward trend, while $SpO_2$ exhibited an overall downward with a slight increase at the fifth minute. The $SpO_2$ nadir for 70 patients (9.3%) was lower than 80%. Further, the $SpO_2$ nadir for 78.27% of the participants appeared at the end of the fourth minute. The 6-min walk distance (6MWD) had the strongest correlation with NYHA classification ($r=0.82$, $p<0.01$). The ratio of 6MWD to predicted 6MWD was most correlated to forced expiratory volume in the first second ($r=0.30$, $p<0.01$) and forced vital capacity ($r=0.30$, $p<0.01$). $SpO_2$ at 3 min had the strongest correlation to patients' diffusing capacity of the lungs for carbon monoxide ($r=0.41$, $p<0.01$). We found significant differences in 6MWD ($F=2.44$, $p=0.033$), $SpO_2$ change ($F=2.58$, $p=0.025$), HR at 0 min ($F=2.87$, $p=0.014$), HR at end of 6 min ($F=2.58$, $p=0.025$) and HR zenith ($F=2.64$, $p=0.022$) between the subtypes of ILD.

**Conclusion** This observation provided an important evidence regarding oxygen titration. It is better to maintain $SpO_2$ above 88% for 4 min instead of 3 min. $SpO_2$ at the third minute was the most valuable predictor of patients' lung function. 6MWD and $SpO_2$ changes were more discriminative in subtypes.

## BACKGROUND

Interstitial lung disease (ILD) is a group of more than 200 kinds of diseases characterised by pulmonary inflammation, accompanied with or without fibrosis.[1 2] Patients diagnosed with ILD mostly have dyspnoea and decreased tolerance to exercise.[3] The 6-min walk test (6MWT) is widely used to assess patients' performance

ability with different cardiopulmonary-related diseases, which provides essential outcomes that cannot be obtained otherwise by standardised pulmonary function testing.[4]

### STRENGTHS AND LIMITATIONS OF THIS STUDY

⇒ The study included a large sample size of patients with interstitial lung disease (ILD) not limited to idiopathic pulmonary fibrosis.
⇒ This was the first study to compare 6-min walk test (6MWT) outcomes between multiple subtypes of ILD.
⇒ This study described the tendency of heart rate and $SpO_2$ minute-by-minute during 6MWT.
⇒ Lack of follow-up hindered the prediction of patients' long-term clinical outcomes using 6MWT.

Until, most 6MWT-related studies have focused on idiopathic pulmonary fibrosis (IPF), which is the most common type of ILD. Previous studies have investigated the outcomes of 6MWT, most of which centred on the 6-min walk distance (6MWD) and percutaneous oxygen saturation ($SpO_2$). However, 6MWT is ideal in predicting patients' clinical outcomes. Previous studies found that 6MWD and oxygen desaturation are associated with mortality in patients with IPF.[5 6] According to a previous study, 6MWD was an independent positive factor for the physical activity of patients with IPF.[7] In addition, 6MWD had a positive association with the subjective health-related quality of life (HRQL) and objective lung function index,[8 9] which included the predicted percentage of forced vital capacity (FVC) and predicted lung diffusing capacity for carbon monoxide (DLco)[5 8 9] as well as forced expiratory volume in the first second ($FEV_1$),[10] which is a negative predictor for dyspnoea.[8 9 11 12] Furthermore, the occurrence of desaturation and changes in $SpO_2$ during the test were indicators of patients' mortality with IPF.[13 14]

6MWT has multiple associated outcomes that are not restricted to 6MWD and $SpO_2$. It comprises

the distance walked, heart rate (HR), blood pressure, $SpO_2$ and dyspnoea, as assessed by the Borg scale.[15] Although the predictive value of oxygen desaturation for the clinical outcomes of the patients has been confirmed,[5 6] the most predictive time point of this outcome within the 6min remains unclear. In addition, the effect of different subtypes of ILD on 6MWT outcomes has not been evaluated yet. Several studies highlighted the importance of finding the most prognostic outcome of 6MWT,[6 16] and measuring $SpO_2$ for the entire 6min duration of 6MWT is recommended by the 2014 technical standards of European Respiratory Society and American Thoracic Society.[17] In the current study, wearable monitors were used to obtain the precise minute-by-minute data of 6MWT, which facilitated the descriptions and comparisons in detail. The comparison between subgroups of ILD will provide new insights into the distinguishing value of 6MWT outcomes.

Hence, to provide detailed features, the predictive value of 6MWT for cardiopulmonary functions, and its distinguishing value for the subtypes of ILD in the current study, we aimed to: (1) describe the detailed outcomes of 6MWT outcomes, including the HR, $SpO_2$, blood pressure, Borg score and walking distance. (2) Identify the correlations between 6MWT outcomes and patients' cardiopulmonary functions. (3) Investigate the effect of the differences between subclassifications of ILD on 6MWT outcomes.

## METHODS

### Design

This was an observational study using a wearable monitor.

### Patients

All the patients were recruited from July 2019 to August 2020 at the Guangzhou Respiratory Health Institute, the biggest respiratory centre in China. We identified eligible participants based on the following inclusion and exclusion criteria—we included patients who were diagnosed with ILD, or whose condition was feasible to conduct 6MWT. The expert pulmonologist established the diagnosis based on patients' symptoms, the radiologist's opinion from the imaging tests, blood tests results, lung function tests, bronchoscopy and biopsy. We excluded patients who had walking limitations, including joint restrictions or other critical diseases and those who experienced myocardial infarction in the previous 5 days, unstable angina, syncope, symptomatic arrhythmia, severe aortic stenosis or decompensated heart failure due to another unstable medical issues.[17] After the initial screening, we obtained informed consent from eligible participants before including them in the study.

### Measurements

#### Demographics questionnaire

The self-designed demographic questionnaire included questions about the age, height, weight, body mass index and sex of the participants.

#### NYHA functional classification

The New York Heart Association (NYHA) classification was considered as a critical criterion for a comprehensive cardiac diagnosis.[18] It classifies patients into four categories, based on their limitations during physical activity, which ranges from no symptoms with ordinary physical activity (class I) to symptoms at rest and increased discomfort with any physical activity (class IV).[19]

#### Borg scale

The Borg Rating of Perceived Exertion scale was developed by Borg,[20] which is widely used to measure patients' effort and exertion, breathlessness and fatigue during physical work. A higher score indicated a more severe level of exertion.[21]

#### Outcomes of 6MWT

According to Enright's recommendations,[22] the primary outcome in our study was 6MWD. We calculated the predicted 6MWD based on equations developed by Enright and Sherrill.[23] For men, the predicted 6MWD=(7.57 * $height_{cm}$)−(5.02 * age)−(1.76 * $weight_{kg}$)−309 m. For women, the predicted 6MWD=(2.11 * $height_{cm}$)−(2.29 * age)−(5.78 * $weight_{kg}$)+667 m. Secondary outcomes include fatigue and dyspnoea, arterial oxygen saturation, HR and blood pressure. We measured fatigue and dyspnoea by modified Borg scale before and after 6MWT, and used wearable monitors to record patients' arterial oxygen saturation and HR during 6MWT. We also recorded the patients' blood pressure before and after the test, and calculate the mean arterial pressure. In this study, 6MWT was conducted without oxygen therapy support. Most of participants had received a 6MWT at the outpatient clinic before their hospital admission. In this case, the learning effect that improves the distance of second walk will be weak.[24] Therefore, we conducted one 6MWT for each patient.

#### Pulmonary function test

Restriction of lung volumes and dysfunction of diffusion are the main functional respiratory abnormalities. An increased $FEV_1$/FVC ratio, accompanied by a low total lung capacity, indicates restriction of lung volumes. Previous studies have proven that reduction in FVC and $DL_{CO}$ are associated with poor survival rates and prognosis.[25] Therefore, in this study, FVC, FEV1 and $DL_{CO}$ were used for the respiratory function assessment.

#### Subtypes of ILD

Since ILD encompasses more than 200 parenchymal pulmonary disorders, we divided all the cases into subtypes, to facilitate the analysis. According to the classifications of Cottin et al,[26] the subtypes contain idiopathic interstitial pneumonias, autoimmune ILDs, hypersensitivity pneumonitis, sarcoidosis and other ILDs. Beause IPF is the most widely studied and the most common type of ILD, we classified it as a dependent category to make the comparisons more detailed. Therefore, we included six subtypes in total.

## Data collection

We collected patients' demographic data using a self-designed questionnaire, which was administered after 6MWT. The outcomes of the pulmonary function test and the NYHA functional classification were obtained from the patients' medical records. The Borg scale was employed before and after walking. All the 6MWT-related outcomes were automatically collected using physiological parameters transmission management software during the 6-min module (Shenzhen zhongruiqi Electronic Technology Co.). Since the patients at the centre completed all the assessments and tests within 3 days of their admission, the outcomes of pulmonary function test, NYHA functional classification and 6MWT were obtained within the next 3 days.

## Analysis

We used the statistical package for the social sciences (SPSS) software V.21.0 (IBM Corporation) for data analysis. We used descriptive statistics to summarise the participants' demographics, 6MWT outcomes, Borg grades, NYHA functional classification and pulmonary function indexes. Specifically, we described continuous variables as mean and SD, and categorical variables as frequency. After performing a check for normality, we used the paired t-test to assess the differences in $SpO_2$ between each end of the minute. Analysis of variance was used to assess the differences of 6MWT-related outcomes across the subtypes of ILD. We performed the Pearson correlation analysis to identify the correlations between the 6MWT outcomes and other measurements. Statistical significance was set at p<0.05.

## Patient and public involvement

No patient involved.

## RESULTS

### Demographics and characteristics of patients

We included 954 patients with ILD from July 2019 to August 2020. The average age of participants was 55.40 (SD=12.35) years (range 14–83 years). The sample included 510 (53.50%) men (table 1).

### Features of 6MWT outcomes among patients with ILD

For 750 participants with valid data, the $SpO_2$ nadir was higher than 80%, and for 524 patients (69.9%) the $SpO_2$ nadir was higher than 88. Other details are shown in table 2. Figure 1 shows patients' $SpO_2$ and HR during 6MWT. $SpO_2$ generally showed a downward trend, but increased slightly at the end of the fifth minute, whereas HR exhibited a sharp increase in the first 2 min and reached a peak before becoming steady. Paired t-test found three significant drops in the $SpO_2$, which occurred at the first minute (t=19.29, p<0.001), the second minute (t=25.38, p<0.001) and the third minute (t=4.75, p<0.001). This was accompanied by a slightly significant rise at the fourth minute (t=−2.06, p=0.039). Figure 2 depicts the time point when

**Table 1** The demographic and characteristic information of patients with interstitial lung disease (ILD) (n=954)

| Variables | Categories | N (%)/Mean (SD) |
| --- | --- | --- |
| Height (cm) | – | 161.08 (8.00) |
| Weight (kg) | – | 62.60 (10.60 |
| BMI | – | 24.07 (3.39) |
| Subclass of ILD | Autoimmune ILDs | 277 (29.00 ) |
| | IIPs | 195 (20.40 ) |
| | IPF | 171 (17.90) |
| | Sarcoidosis | 50 (5.20 ) |
| | Hypersensitivity pneumonitis | 177 (18.60 ) |
| | Others ILDs | 37 (3.90 ) |
| | Missing data | 47 (4.90 ) |

BMI, body mass index; IIPs, Idiopathic interstitial pneumonias; IPF, idiopathic pulmonary fibrosis.

$SpO_2$ nadir appears at the first time and the occurrence of $SpO_2$ nadir at each end of a minute over 6 min. The $SpO_2$ nadir of 63.87% and 78.27% of the participants' appeared at the end of the third and the fourth minute, respectively.

### Difference in 6MWT outcomes among subgroup of patients with ILD

Significant differences between the subtypes of ILD were found for 6MWD (F=2.44, p=0.033), $SpO_2$ change (F=2.58, p=0.025), HR at 0 min (F=2.87, p=0.014), HR at end of 6 min (F=2.58, p=0.025) and HR zenith (F=2.64, p=0.022) (table 3).

### Correlation between the outcomes of 6MWT and cardiopulmonary function

$SpO_2$ was generally positively correlated to cardiopulmonary function, whereas the HR and Borg scale were negatively correlated. Specifically, the NYHA grade strongly correlated with 6MWD (r=0.82, p<0.01). The 6MWD/predicted 6MWD had the highest correlation coefficient with FVC (r=0.30, p<0.01) and $FEV_1$ (r=0.30, p<0.01). $SpO_2$ at the end of 3 min had the strongest correlation to $DL_{CO}$ (r=0.41, p<0.01) (table 4).

## DISCUSSION

This study described identified the correlation between 6MWT outcomes and cardiopulmonary function and compared the difference between subtypes of ILD on 6MWT outcomes. We found that the HR and $SpO_2$ did not increase or decrease uniformly during walking. For approximately 10% of the patients, the $SpO_2$ nadir was lower than 80%, but they completed the test. Besides, $SpO_2$ nadir appeared at the end of the fourth minute for approximately 80% of patients. Therefore, 6MWD and $SpO_2$ had the strongest correlation with heart function and lung function of ILD, respectively. Moreover,

**Table 2** The features of 6-min walk test (6MWT) outcomes among patients with interstitial lung disease

| Items | N | Minimum | Maximum | Mean (SD) |
|---|---|---|---|---|
| Systolic blood pressure before 6MWT (mm Hg) | 950 | 84 | 188 | 124.16 (17.26) |
| Systolic blood pressure after 6MWT (mm Hg) | 734 | 88 | 242 | 138.46 (23.19) |
| Diastolic blood pressure before 6MWT (mm Hg) | 950 | 50 | 128 | 77.64 (11.58) |
| Diastolic blood pressure after 6MWT (mm Hg) | 734 | 49 | 167 | 82.45 (13.44) |
| Mean arterial pressure before 6MWT (mm Hg) | 950 | 62.67 | 148.00 | 93.15 (12.16) |
| Mean arterial pressure before 6MWT (mm Hg) | 734 | 66.67 | 182.00 | 101.12 (14.89) |
| Heart rate at 0 min (times/min) | 750 | 50 | 141 | 90.97 (14.48) |
| Heart rate at 1 min (times/min) | 749 | 65 | 177 | 107.69 (15.25) |
| Heart rate at 2 min (times/min) | 749 | 70 | 192 | 113.66 (16.33) |
| Heart rate at 3 min (times/min) | 749 | 68 | 199 | 115.43 (16.88) |
| Heart rate at 4 min (times/min) | 749 | 70 | 201 | 116.08 (17.65) |
| Heart rate at 5 min (times/min) | 749 | 66 | 195 | 116.49 (17.77) |
| Heart rate at 6 min (times/min) | 749 | 66 | 193 | 117.97 (18.00) |
| Heart rate zenith (times/min) | 750 | 70 | 201 | 121.38 (19.18) |
| Heart rate change (times/min) | 749 | −102.00 | 29.00 | −27.07 (14.45) |
| $SpO_2$ at 0 min (%) | 750 | 82 | 100 | 95.49 (2.23) |
| $SpO_2$ at 1 min (%) | 748 | 74 | 100 | 93.61 (3.46) |
| $SpO_2$ at 2 min (%) | 749 | 65 | 99 | 91.03 (4.99) |
| $SpO_2$ at 3 min (%) | 749 | 65 | 99 | 90.62 (5.82) |
| $SpO_2$ at 4 min (%) | 749 | 58 | 99 | 90.49 (6.13) |
| $SpO_2$ at 5 min (%) | 749 | 61 | 99 | 90.65 (6.23) |
| $SpO_2$ at 6 min (%) | 748 | 56 | 99 | 90.54 (6.43) |
| $SpO_2$ nadir (%) | 750 | 56.00 | 99.00 | 89.09 (6.44) |
| $SpO_2$ change (%) | 748 | −10.00 | 33.00 | 4.96 (5.57) |
| Distance (m) | 953 | 53 | 999 | 457.28 (98.40) |
| Distance/ predicted distance (m) | 953 | 12.49 | 175.1 | 84.74 (18.54) |

Heart rate change and $SpO_2$ change were the values at the beginning minus the values at the end of 6 min respectively.
$SpO_2$, peripheral capillary oxygen saturation.

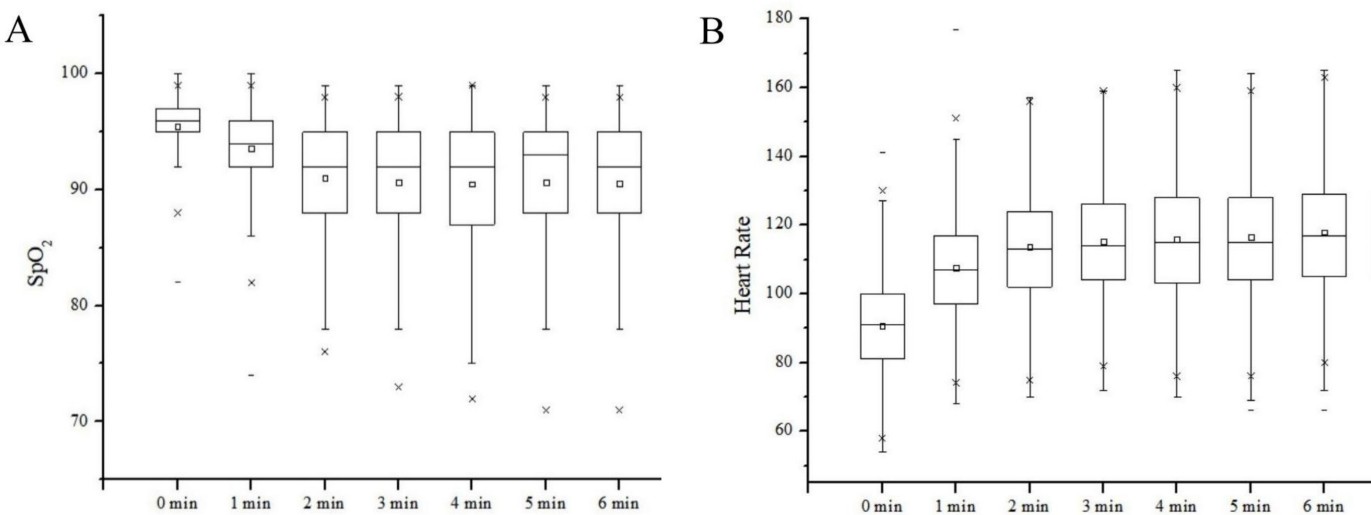

**Figure 1** Description of SPO$_2$ and heart rate (HR) in patients with interstitial lung disease during 6-min walk test.

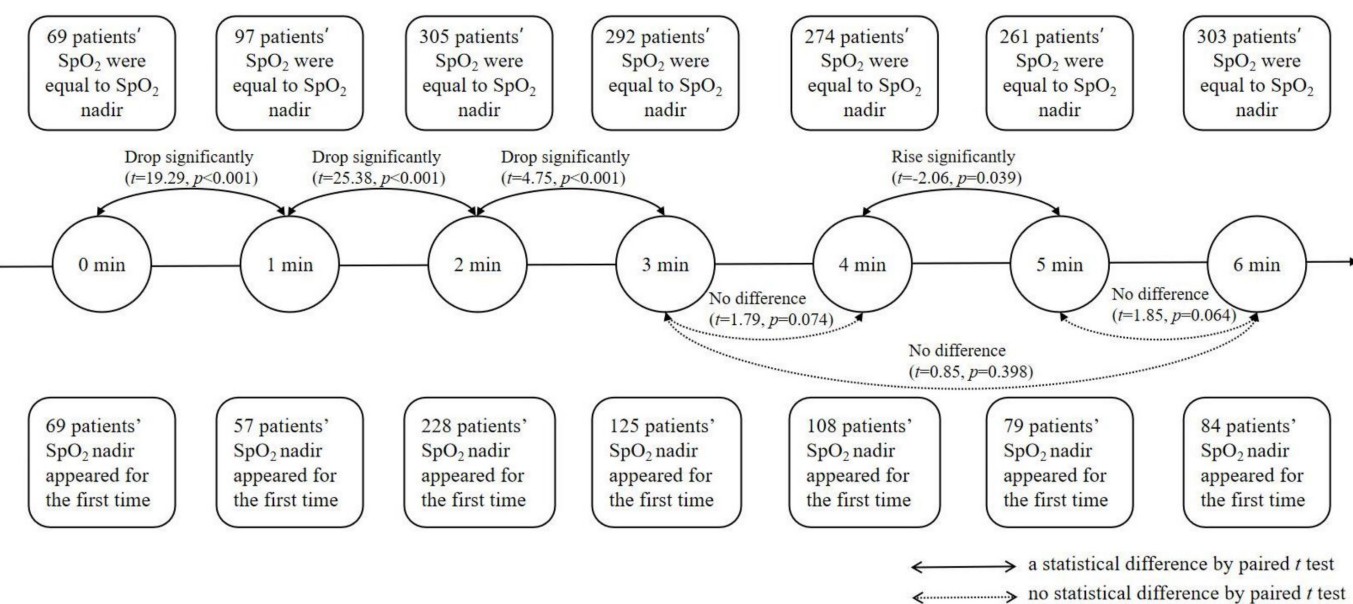

**Figure 2** Statistical analysis chart of $SpO_2$ in patients with interstitial lung disease within 6 min.

group comparisons revealed that the 6MWD and $SpO_2$ change were more distinguishing for the subgroups of ILD.

Compared with previous studies, the average 6MWD in our study was 457.28 m, which was moderate.[27 28] HR increased continuously and $SpO_2$ decreased, with a slight rise in the fifth minute. However, the results of a previous study showed a slight increase in the fourth minute and a sharp drop in $SpO_2$.[29] Since our study had a bigger sample size, the average 6MWD and tendency of $SpO_2$ were more representative. According to the standard, 6MWT should be terminated when $SpO_2$ falls below 80%.[30] When $SpO_2$ was less than 88%, it was considered as a significant desaturation, and patients were recommended to take an oxygen supplement.[31–33] Without oxygen supplements in our study, $SpO_2$ nadir were lower than 88% and 80% for 30.1% and 9.3% of the patients, respectively. They all completed 6MWT without any chest pain, leg cramps, unsteady gait, diaphoresis or a pale/ashen appearance, experiencing breathlessness, or reporting being too tired to continue. Our findings indicated that it is unwarranted to stop 6MWT when patients with ILD only experience desaturation without other indications of termination, which corroborate the findings of Afzal et al.[34] The $SpO_2$ nadir is an essential outcome of 6MWT, and our research revealed that for 63.87% and 78.27% of the participants, $SpO_2$ nadir appeared at the end of the third and fourth minute respectively. Oxygen titration is generally performed with 6MWT to determine the oxygen flow that prevents oxygen saturation from falling below 88%, measured using pulse oximetry ($SpO_2$). According to Giovacchini et al,[35] after a certain dose of oxygen is administered, the patients' $SpO_2$ should exceed 88% and be stable for 3 min. In our study, we found that the $SpO_2$ nadir for approximately 80% of the patients had appeared at the end of the fourth minute; hence, we

strongly recommend that oxygen titration should be for 4 min.

Garin et al did not find significant differences between IPF and systemic sclerosis-associated ILD on 6MWD and dyspnoea,[36] while Someya and Mugii found that patients with IPF had lower $SpO_2$ and higher Borg score than patients with dermatomyositis.[37] We observed no significant differences between the subtypes for Borg score and $SpO_2$ after walking. Since previous studies merely compared two different subgroups of ILD, our results were more comprehensive and reliable. Contrary to dyspnoea and $SpO_2$ after walking, we found significant differences between groups on 6MWD and $SpO_2$ change. Therefore, 6MWD and $SpO_2$ change was the more distinguishing outcomes for subtypes of ILD. Although the HR at the 0 min, end of 6 min and HR zenith showed significant differences between the subtypes, this finding was unclear because baseline HR showed differences before walking. Therefore, future studies in another population or multicentre may reinforce the findings.

Similar to previous studies, 6MWD and $SpO_2$ positively correlated with cardiopulmonary function outcomes such as NYHA, FVC, $FEV_1$ and $DL_{CO}$,[5 8–10] while Borg score was negatively correlated.[30] Compared with 6MWD and $SpO_2$, the patients' HR had a weaker positive correlation to cardiopulmonary function outcomes. Hence, $SpO_2$ and distance were more valuable than HR in predicting the patients' cardiopulmonary function and degree of dyspnoea. In a study, 6MWD was more correlated to $DL_{CO}$ than $SpO_2$,[27] however, our result was to the contrary—lower FVC and $DL_{CO}$ were associated with poor prognosis and high mortality.[16 38–40] Nevertheless, the $DL_{CO}$ level was more valuable than FVC, as it captured the combined impact on the pulmonary reserve of IPF, emphysema and pulmonary hypertension.[41] Since $SpO_2$ had the strongest correlation with $DL_{CO}$, we recommend

**Table 3** Difference between subgroups of interstitial lung disease (ILD) in 6-min walk test (6MWT) outcomes

| Measures | Categories | N | Mean(SD) | F | P value |
|---|---|---|---|---|---|
| Borg score before 6MWT | Autoimmune ILDs | 267 | 0.25 (0.54) | 1.59 | 0.16 |
| | IIPs | 192 | 0.18 (0.50) | | |
| | IPF | 167 | 0.28 (0.51) | | |
| | Sarcoidosis | 49 | 0.18 (0.39) | | |
| | Others ILDs | 37 | 0.35 (0.63) | | |
| | Hypersensitivity pneumonitis | 174 | 0.31 (0.59) | | |
| Borg score after 6MWT | Autoimmune ILDs | 267 | 1.25 (1.15) | 0.97 | 0.433 |
| | IIPs | 192 | 1.32 (1.21) | | |
| | IPF | 167 | 1.37 (1.23) | | |
| | Sarcoidosis | 49 | 1.04 (0.98) | | |
| | Others ILDs | 36 | 1.14 (1.17) | | |
| | Hypersensitivity pneumonitis | 174 | 1.38 (1.15) | | |
| 6MWD (m) | Autoimmune ILDs | 277 | 452.69 (96.25) | 2.44 | 0.033 |
| | IIPs | 195 | 466.07 (96.86) | | |
| | IPF | 171 | 440.70 (87.85) | | |
| | Sarcoidosis | 50 | 479.36 (91.25) | | |
| | Others ILDs | 37 | 482.30 (116.70) | | |
| | Hypersensitivity pneumonitis | 177 | 453.48 (107.31) | | |
| 6MWD/predicted 6MWD | Autoimmune ILDs | 277 | 0.84 (0.18) | 0.84 | 0.521 |
| | IIPs | 195 | 0.85 (0.19) | | |
| | IPF | 171 | 0.87 (0.18) | | |
| | Sarcoidosis | 50 | 0.87 (0.17) | | |
| | Others ILDs | 37 | 0.84 (0.23) | | |
| | Hypersensitivity pneumonitis | 177 | 0.84 (0.19) | | |
| SpO2 at the 0 min (%) | Autoimmune ILDs | 219 | 95.75 (1.98) | 2.14 | 0.059 |
| | IIPs | 143 | 95.15 (2.53) | | |
| | IPF | 143 | 95.14 (2.33) | | |
| | Sarcoidosis | 45 | 95.40 (1.76) | | |
| | Others ILDs | 27 | 95.37 (3.12) | | |
| | Hypersensitivity pneumonitis | 135 | 95.70 (2.21) | | |
| $SpO_2$ at the end of 6 min (%) | Autoimmune ILDs | 218 | 90.78 (6.07) | 2.14 | 0.059 |
| | IIPs | 143 | 91.07 (5.74) | | |
| | IPF | 143 | 88.91 (6.49) | | |
| | Sarcoidosis | 45 | 90.89 (6.41) | | |
| | Others ILDs | 27 | 91.07 (5.86) | | |
| | Hypersensitivity pneumonitis | 134 | 90.07 (7.85) | | |
| SpO2 nadir (%) | Autoimmune ILDs | 219 | 89.20 (6.01) | 1.421 | 0.215 |
| | IIPs | 143 | 89.60 (5.52) | | |
| | IPF | 143 | 87.86 (6.56) | | |
| | Sarcoidosis | 45 | 88.89 (6.47) | | |
| | Others ILDs | 27 | 90.15 (5.82) | | |
| | Hypersensitivity pneumonitis | 135 | 88.59 (8.08) | | |
| $SpO_2$ change (%) | Autoimmune ILDs | 218 | 4.98 (5.52) | 2.58 | 0.025 |
| | IIPs | 143 | 4.08 (5.04) | | |
| | IPF | 143 | 6.23 (5.05) | | |
| | Sarcoidosis | 45 | 4.51 (5.88) | | |
| | Others ILDs | 27 | 4.30 (3.61) | | |

**Table 3** Continued

| Measures | Categories | N | Mean(SD) | F | P value |
|---|---|---|---|---|---|
| | Hypersensitivity pneumonitis | 134 | 5.63 (6.93) | | |
| HR at the 0 min (times/min) | Autoimmune ILDs | 219 | 92.21 (15.09) | 2.87 | 0.014 |
| | IIPs | 143 | 91.12 (12.75) | | |
| | IPF | 143 | 87.15 (13.80) | | |
| | Sarcoidosis | 45 | 88.44 (14.46) | | |
| | Others ILDs | 27 | 91.81 (13.76) | | |
| | Hypersensitivity pneumonitis | 135 | 92.40 (15.61) | | |
| HR at the end of 6 min (times/min) | Autoimmune ILDs | 219 | 119.45 (17.65) | 2.58 | 0.025 |
| | IIPs | 143 | 118.76 (18.63) | | |
| | IPF | 143 | 113.05 (15.41) | | |
| | Sarcoidosis | 45 | 118.33 (20.99) | | |
| | Others ILDs | 27 | 116.30 (16.22) | | |
| | Hypersensitivity pneumonitis | 135 | 118.57 (18.77) | | |
| HR zenith (times/min) | Autoimmune ILDs | 219 | 122.29 (18.61) | 2.64 | 0.022 |
| | IIPs | 143 | 121.86 (19.58) | | |
| | IPF | 143 | 115.90 (15.79) | | |
| | Sarcoidosis | 45 | 124.09 (23.88) | | |
| | Others ILDs | 27 | 122.22 (19.73) | | |
| | Hypersensitivity pneumonitis | 135 | 122.07 (20.53) | | |
| HR change (times/min) | Autoimmune ILDs | 219 | −27.23 (14.95) | 0.8 | 0.547 |
| | IIPs | 143 | −27.64 (15.23) | | |
| | IPF | 143 | −25.90 (12.03) | | |
| | Sarcoidosis | 45 | −29.89 (14.32) | | |
| | Others ILDs | 27 | −24.48 (12.92) | | |
| | Hypersensitivity pneumonitis | 134 | −26.54 (14.52) | | |

HR, heart rate; IIPs, Idiopathic interstitial pneumonias; IPF, idiopathic pulmonary fibrosis; 6MWD, 6-min walk distance; $SpO_2$, peripheral capillary oxygen saturation.

clinical practitioners to monitor the $SpO_2$ of patients with ILD. Besides, the most valuable $SpO_2$ time point remains unclear. A previous study revealed that $SpO_2$ nadir and $SpO_2$ change had the same degree of correlation with $DL_{CO}$,[42] and another study also highlighted the critical predictive value of the $SpO_2$ nadir.[43] In contrast, our findings illustrated that $SpO_2$ at the end of the third minute was more predictive than the $SpO_2$ nadir and $SpO_2$ change in $DL_{CO}$. Although 6MWD/predicted 6MWD had a higher correlation to $FEV_1$ and FVC than $SpO_2$ at the end of the third minute, 6MWD was more susceptible to factors such as age, sex, shorter corridor and inappropriate walking shoes.[44–46] Furthermore, $DL_{CO}$ was considered more critical than $FEV_1$ and FVC for ILD. Hence, the third-minute $SpO_2$ can be an alternative to predict lung function in patients with ILD.

### Limitations and future research directions

This study had several limitations. First, we did not conduct the second 6MWT for patients, and so the measured distance might not be the longest potential distance. Second, 25% of the values regarding some non-critical variables were missing, which might introduce selection bias and affect the validity and representativeness of the results. Replication in another population or a multicentric study could reinforce the findings. Third, lack of follow-up on patients' prognosis and mortality hindered the prediction of 6MWT outcomes on the long-term clinical outcomes. Future research is required to explore the association between the outcomes of 6MWT and long-term prognosis.

### CONCLUSIONS

Despite the above limitations, this study showed that increased HR and decreased $SpO_2$ during the 6MWT do not change uniformly. Approximately 10% of the patients, whose $SpO_2$ was less than 80%, completed 6MWT without any discomfort indicated. Hence, it is unwarranted to halt 6MWT when patients with ILD experience only desaturation, without other indications of termination. $SpO_2$ nadir appeared at the end of the fourth minute for approximately 80% of the patients, which provides an important

**Table 4** Correlations between 6-min walk test (6MWT) outcomes and cardiopulmonary function

| Outcomes of 6MWT | | NYHA | MAP before 6MWT | MAP after 6MWT | MAP change | FVC | FEV$_1$ | DL$_{co}$ |
|---|---|---|---|---|---|---|---|---|
| 6MWD (m) | r | **0.82**\*\* | 0.07\* | 0.09\* | 0.08\* | 0.24\*\* | 0.17\*\* | 0.26\*\* |
| | n | 751 | 949 | 733 | 733 | 846 | 846 | 806 |
| 6MWD/predicted 6MWD | r | 0.64\*\* | 0.09\*\* | 0.14\*\* | 0.09\* | **0.30**\*\* | **0.30**\*\* | 0.28\*\* |
| | n | 751 | 949 | 733 | 733 | 846 | 846 | 806 |
| SpO$_2$ at 0 min (%) | r | 0.29\*\* | 0.03 | 0.01 | −0.02 | 0.18\*\* | 0.17\*\* | 0.26\*\* |
| | n | 750 | 746 | 732 | 732 | 644 | 644 | 604 |
| SpO$_2$ at end of 1 min (%) | r | 0.27\*\* | 0.02 | 0.01 | 0.01 | 0.24\*\* | 0.21\*\* | 0.31\*\* |
| | n | 748 | 744 | 730 | 730 | 642 | 642 | 602 |
| SpO$_2$ at end of 2 min (%) | r | 0.28\*\* | 0.05 | 0.03 | −0.01 | 0.27\*\* | .23\*\* | .37\*\* |
| | n | 749 | 745 | 731 | 731 | 643 | 643 | 603 |
| SpO$_2$ at end of 3 min (%) | r | 0.29\*\* | 0.047 | 0.034 | 0.00 | **0.29**\*\* | **0.24**\*\* | **0.41**\*\* |
| | n | 749 | 745 | 731 | 731 | 643 | 643 | 603 |
| SpO$_2$ at end of 4 min (%) | r | 0.27\*\* | 0.03 | 0.03 | 0.01 | 0.27\*\* | 0.23\*\* | 0.40\*\* |
| | n | 749 | 745 | 731 | 731 | 643 | 643 | 603 |
| SpO$_2$ at end of 5 min (%) | r | 0.27\*\* | 0.03 | 0.03 | 0.01 | 0.27\*\* | 0.23\*\* | 0.39\*\* |
| | n | 749 | 745 | 731 | 731 | 643 | 643 | 603 |
| SpO$_2$ at end of 6 min (%) | r | 0.26\*\* | 0.03 | 0.02 | 0.01 | 0.27\*\* | 0.22\*\* | 0.37\*\* |
| | n | 748 | 744 | 730 | 730 | 642 | 642 | 602 |
| SpO$_2$ nadir (%) | r | 0.27\*\* | 0.05 | 0.04 | 0.01 | 0.28\*\* | 0.24\*\* | 0.38\*\* |
| | n | 750 | 746 | 732 | 732 | 644 | 644 | 604 |
| SpO$_2$ change (%) | r | −0.18\*\* | −0.02 | −0.02 | −0.02 | −0.24\*\* | −0.19\*\* | −0.32\*\* |
| | n | 748 | 744 | 730 | 730 | 642 | 642 | 602 |
| Borg scale at 0 min | r | −0.23\*\* | −0.02 | 0.02 | 0.05 | −0.12\*\* | −0.06 | −0.13\*\* |
| | n | 727 | 926 | 714 | 714 | 829 | 829 | 791 |
| Borg scale at the end | r | −0.25\*\* | −0.05 | 0.00 | 0.07 | −0.15\*\* | −0.11\*\* | −0.19\*\* |
| | n | 726 | 925 | 713 | 713 | 828 | 828 | 790 |
| HR at 0 min (times/min) | r | −0.02 | 0.05 | 0.03 | −0.01 | −0.18\*\* | −0.21\*\* | −0.10\* |
| | n | 750 | 746 | 732 | 732 | 644 | 644 | 604 |
| HR at end of 1 min (times/min) | r | 0.16\*\* | 0.09\*\* | 0.106\*\* | 0.04 | −0.08 | −0.12\*\* | 0.00 |
| | n | 749 | 745 | 731 | 731 | 643 | 643 | 603 |
| HR at end of 2 min (times/min) | r | 0.18\*\* | 0.09\* | 0.13\* | 0.07 | −0.08\* | −0.11\*\* | −0.02 |
| | n | 749 | 745 | 731 | 731 | 643 | 643 | 603 |
| HR at end of 3 min (times/min) | r | 0.18\*\* | 0.12\*\* | 0.16\*\* | 0.09\* | −0.08 | −0.11\*\* | −0.01 |
| | n | 749 | 745 | 731 | 731 | 643 | 643 | 603 |
| HR at end of 4 min (times/min) | r | 0.19\*\* | 0.10\*\* | 0.14\*\* | 0.08\* | −0.06 | −0.11\*\* | −0.03 |
| | n | 749 | 745 | 731 | 731 | 643 | 643 | 603 |
| HR at end of 5 min (times/min) | r | 0.22\*\* | 0.11\*\* | 0.16\*\* | 0.10\*\* | −0.05 | −0.09\* | −0.02 |
| | n | 749 | 745 | 731 | 731 | 643 | 643 | 603 |
| HR at end of 6 min (times/min) | r | 0.24\*\* | 0.10\*\* | 0.16\*\* | 0.11\*\* | −0.04 | −0.09\* | −0.02 |
| | n | 749 | 745 | 731 | 731 | 643 | 643 | 603 |
| HR zenith (times/min) | r | 0.19\*\* | 0.10\*\* | 0.12\*\* | 0.06 | −0.03 | −0.09\* | 0.01 |
| | n | 750 | 746 | 732 | 732 | 644 | 644 | 604 |
| HR change (times/min) | r | −0.32\*\* | −0.08\* | −0.17\* | −0.15\*\* | −0.14\*\* | −0.11\*\* | −0.09\* |
| | n | 749 | 745 | 731 | 731 | 643 | 643 | 603 |

\*\*P<0.01, \*p<0.05.
DLCO, diffusing capacity of the lungs for carbon monoxide; FEV$_1$, forced expiratory volume in the first second; FVC, forced vital capacity; HR, heart rate; MAP, mean arterial pressure; 6MWD, 6-min walk distance; NYHA, New York Heart Association; SpO$_2$, peripheral capillary oxygen saturation.

evidence regarding oxygen titration, that is, it is better to maintain $SpO_2$ above 88% for 4 min. Besides, the third-minute $SpO_2$ can be an alternative to predict patients' lung function. Conclusively, 6MWD and $SpO_2$ change showed significant differences between the subtypes of ILD, which indicated that they were more distinguishing for the subtypes of ILD.

**Acknowledgements** The authors thank the other medical works for facilitating our data collection, they are Yanni XU, Yanfen PAN, Yanqiu LV and Xian LI. We would also like to thank Editage (www.editage.cn) for English language editing.

**Contributors** JL and XW were involved in the design of the study. MD, Xiaoyan L and HW were responsible for the data collection. Xinyin L was involved in data analysis. JL drafted the manuscript. All the authors made contributions to the revision and approved the final version. XW is responsible for the overall content as guarantor

**Funding** This research was supported by the First Affiliated Hospital of Guangzhou Medical University, and the funding number is ZH201822.

**Competing interests** None declared.

**Patient and public involvement** Patients and/or the public were not involved in the design, or conduct, or reporting, or dissemination plans of this research.

**Patient consent for publication** Consent obtained directly from patient(s)

**Ethics approval** Ethical principles of the Declaration of Helsinki was used for guide this study, and the research protocol has been approved by the ethics committee of the First Affiliated Hospital of Guangzhou Medical University. The approval number is 2020 No.K-45. Participants gave informed consent to participate in the study before taking part.

**Provenance and peer review** Not commissioned; externally peer reviewed.

**Data availability statement** Data are available upon reasonable request. Data were available upon reasonable request from Xiaobing Wu (wuxiaobing_gz@163.com)

**ORCID iD**
Xiaobing Wu http://orcid.org/0000-0003-4644-2325

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
