## [Reviewer comments · BMJ Open]

ARTICLE DETAILS

TITLE (PROVISIONAL)	Features and Predictive Value of Six-minute Walk Test Outcomes in Interstitial Lung Disease: An Observation Study using Wearable Monitors
AUTHORS	Li, Jiaying; LI, Xiaoyan; DENG, Miaozen; LIANG, Xinyin; WEI, Huiqun; Wu, Xiaobing

VERSION 1 – REVIEW

REVIEWER	Ayed, Khadija University of Tunis El Manar, physiologie
REVIEW RETURNED	09-Sep-2021

GENERAL COMMENTS	Corrections are indicated between brackets This is an interesting study which confirms the crucial importance of the six-minute walk test findings in the functional assessment of ILD English writing needs to be revised Methods : what reference did you use to calculate the expected walking distance ? Discussion : in interpreting the correlations, you did not discuss the relationship between the results of the six-minute walk test and blood pressure figures Page 6 ; line 4 : modified Borg (modified Borg scale) Page 6 ; line 11-12 : Restricted ventilation dysfunction and diffusion dysfunction are the main symptoms of interstitial lung disease (restriction of lung volumes and diffusion dysfunction are the main functional respiratory abnormalities) Page 6 ; line 13 : (the FEV1/FVC ratio can be increased in the restriction of lung volumes but this deficit is only confirmed by a TLC below the lower limit of normal) Page 6 ; line 19-21 : it is better to replace this sentence with this (Therefore, in this study FVC, forced expiratory volume in one second (FEV1) and DLCO are used in the respiratory function assessment) Page 6 ; line 37-39: in carrying out the six-minute walk test, it is recommended by learned societies to carry out two successive tests to satisfy the reproducibility of the test Table 1 : you always need the meaning of the abbreviations below the table Page 8 ; line 11-13 : and 524 patients (69.9%) has (have) a higher SpO2 nadir than 88
---

	Page 11 ; line 49-50 : correlated to and cardiopulmonary function (correlated to) Page 13 ; Line :45-47 : SpO2 less than 88% during exercise is considered as desaturation significantly (is considered as a significant desaturation) and be recommended to given oxygen supplement
--	--

REVIEWER	Bertsias, Giorgos University of Crete
REVIEW RETURNED	05-Feb-2022

GENERAL COMMENTS	Unfortunately, the paper is poorly written, it contains numerous syntax errors which make it difficult to read and comprehend.  - On several occasions, language is inappropriate for the a research draft such as for example, "This study is by far the most detailed description of the results of...", "As we all know, the 6MWT..." etc. - The methodology is poorly described, ie. how patients were enrolled, how diagnosis was established, etc. - the novelty of the work is not clear. For me the most clinically relevant point is that "third-minute SpO2 of 6MWT can be an alternative to predict patients' prognosis". However, this is not clearly spelled out in the manuscript. - There were many missing data (up to 25% for some parameters), which may impact on the validity and generalisability of the results. - I am not sure about the relevance of the ILD classification. Within each group (e.g autoimmune ILD), multiple subtypes may exist (NSIP, UIP, LIP etc) which might (or, might not!) have different effect of cardiorespiratory indices.
---

REVIEWER	Juge, Pierre-Antoine Hopital Bichat - Claude-Bernard
REVIEW RETURNED	28-Feb-2022

GENERAL COMMENTS	In this observational study, authors retrospectively collected 6MWT from patients with ILD of various cause in order to validate the 6MWT outside IPF and to investigate the correlation between the variables of the 6MWT and the cardiopulmonary function (NYHA scale, FVC, FEV, DLCO). If the 6MWT is already used in ILD outside IPF, only literature have investigated this test in ILD outside IPF. The great number of included patients (680) having various causes of ILD makes it original. As authors stated, the main limitation of this study is the lack of investigation of an association with prognosis/mortality. Here are my comments: MAJOR COMMENTS  1. Authors have conducted this study because of the lack of data about 6WT in ILD outside IPF. They used the classification of Cottin et al in order to classify ILD causes. I think it would be useful to separate IPF from the other idiopathic ILD and analyze them separately. 2. Distance was measured using meters. However, it varies widely according to age and sex. Authors could transform the data to %of expected in a same sex/same age population using official references. 3. Authors give the results from heart rate and SpO2 at 6 min and at the end of the test. What is the difference between this 3
---

	measures? When was the measure at the end of the test assessed? 4. Methods: Please specify included time and location of the included patients (multicenter? Monocenter?). The methods should specify how diagnosis was performed (expert pulmonologist? Radiologist involved?). 5. Could the authors specify when the PFTs were performed and the NYHA scale collected regarding the 6MWT? In another way, was the PFTs performed the same day than the 6MWT? Same comment for NYHA data. 6. Limitations: replication in another population (or multicentric study) could have reinforce the message. Please comment in the discussion. MINOR COMMENTS 1. Abstract: please rephrase the objective. It does not seems to me that it reflect well the main objective of the study (investigate the 6MWT and its variable among ILD in and outside IPF and their correlation with NYHA and PFT). 2. I would have move Table 3 in supplementary 3. Table1 and 4. Please specify the unit used for each variable. Female (or Male) sex can be removed from the table (1 is sufficient because the other become implicit) 4. Please change Gender to Sex 5. Please have the manuscript English rewrite. Overall, the manuscript is well written but there are small mistakes (past/present tense, missing words, hard to understand phrasing...)
--	--

VERSION 1 – AUTHOR RESPONSE

Reviewer: 1 Dr. Khadija Ayed, University of Tunis El Manar

Dear Dr. Khadija Ayed, thanks for your precious time dedicated to reviewing our paper. To facilitate your further review, we marked the changes in blue in the manuscript.

Comments to the Author:

1. Corrections are indicated between brackets. This is an interesting study that confirms the crucial importance of the six-minute walk test findings in the functional assessment of ILD

Response to the reviewer: We appreciate your positive feedback, and valuable suggestions here.

2. English writing needs to be revised

Response to the reviewer: We have improved the quality of writing throughout the manuscript with the help of a professional editing service.

3. Methods: what reference did you use to calculate the expected walking distance?

Response to the reviewer: Thanks. We have supplemented the equations and the reference we adopted for the predicted six-minute walk distance calculations.

4. Discussion: in interpreting the correlations, you did not discuss the relationship between the results of the six-minute walk test and blood pressure figures

Response to the reviewer: Thanks for your suggestions. We have supplemented the correlations between outcomes of 6MWT and blood pressure figures, including mean arterial pressure before 6MWT minus mean arterial pressure after 6MWT and changes of mean arterial pressure. New results have been added in Table 4, and related changes have been made in the methods section.

5. Page 6 ; line 4 : modified Borg (modified Borg scale)

Response to the reviewer: Thank you for reminding this typo. We have revised it.

6. Page 6 ; line 11-12 : Restricted ventilation dysfunction and diffusion dysfunction are the main symptoms of interstitial lung disease (restriction of lung volumes and diffusion dysfunction are the main functional respiratory abnormalities)

Response to the reviewer: We have revised it in the paper. Changes have been made in

7. Page 6 ; line 13 : (the FEV1/FVC ratio can be increased in the restriction of lung volumes but this deficit is only confirmed by a TLC below the lower limit of normal)

Response to the reviewer: Thank you for pointing out this. We have revised the sentence to make it more clear. Changes have been made in the "Pulmonary function test" section.

8. Page 6 ; line 19-21 : it is better to replace this sentence with this (Therefore, in this study FVC, forced expiratory volume in one second (FEV1) and DLCO are used in the respiratory function assessment)

Response to the reviewer: Thank you for polishing our sentence. We have revised it. Changes have been made in the "Pulmonary function test" section.

9. Page 6 ; line 37-39: in carrying out the six-minute walk test, it is recommended by learned societies to carry out two successive tests to satisfy the reproducibility of the test

Response to the reviewer: Thanks for your valuable comments here. We agree that it is one of the limitations of our study, which will be mentioned in the limitation section. In this study, the patients all come from the respiratory center. Before admission, they usually received a 6MWT in outpatient. Since most of them are not new to 6 minute walk test, the learning effect on improving the 6 minute walk distance in the second time would be weakened. We explained this in the methods section and emphasized it as a limitation for the readers. Changes have been made in the methods and limitations sections.

10. Table 1 : you always need the meaning of the abbreviations below the table

Response to the reviewer: We have annotated the abbreviations and their expansions below the table. We also supplemented that for all tables. Changes have been made below each table.

11. Page 8 ; line 11-13 : and 524 patients (69.9%) has (have) a higher SpO₂ nadir than 88

Response to the reviewer: Thank you for polishing our sentence, and we have revised it

12. Page 11 ; line 49-50 : correlated to and cardiopulmonary function (correlated to)

Response to the reviewer: We have revised the sentence based on your advice.

13. Page 13 ; Line :45-47 : SpO₂ less than 88% during exercise is considered as desaturation significantly (is considered as a significant desaturation) and be recommended to given oxygen supplement

Response to the reviewer: We have revised the sentence according to your suggestion.

Reviewer: 2

Dr. Giorgos Bertias, University of Crete

Dear Dr. Giorgos Bertias, thanks for reviewing our manuscript and your comments. We have made changes based on your suggestions. The revised contents were marked blue in the manuscript.

Comments to the Author:

1. Unfortunately, the paper is poorly written. It contains numerous syntax errors, which make it difficult to read and comprehend.

On several occasions, language is inappropriate for the research draft, such as "This study is by far the most detailed description of the results of...", "As we all know, the 6MWT..." etc.

Response to the reviewer: Thanks for your comment. We have improved our writing of the manuscript with the help of a professional editing service and hope this version will meet your standards.

2. The methodology is poorly described, ie. how patients were enrolled, how the diagnosis was established, etc.

Response to the reviewer: Thanks. We have revised the methods section and added information about the enrollment process and establishment of diagnosis. We also made revisions to elaborate on other aspects of methods in detail. Changes have been made in the whole methods section.

3. the novelty of the work is not clear. For me the most clinically relevant point is that "third-minute SpO₂ of 6MWT can be an alternative to predict patients' prognosis". However, this is not clearly spelled out in the manuscript.

Response to the reviewer: Based on your suggestions, we have emphasized the novelty of this paper in the abstract, discussion, and conclusion sections. Changes were marked in blue.

4. There were many missing data (up to 25% for some parameters), which may impact the validity and generalisability of the results.

Response to the reviewer: We agree with the reviewer's comment. Although the large sample size is one of the strengths of this study, the proportion of missing values on several non-core variables was up to 25%. However, the critical variable in this study is 6-minute walk distance, which is less than 5% missing data. Nevertheless, we mentioned this in the limitation section to remind readers of the interpretations of the related results. Changes have been made in the limitations section.

5. I am not sure about the relevance of the ILD classification. Within each group (e.g autoimmune ILD), multiple subtypes may exist (NSIP, UIP, LIP etc) which might (or, might not!) have different effect of cardiorespiratory indices.

Response to the reviewer: The term interstitial lung disease (ILD) encompasses a large group of > 200 parenchymal pulmonary disorders, of which the majority are classified as rare. As the example you pointed out, there are sub-types under each category. This makes it difficult to compare all the types. As a result, we roughly divided the ILD into sub-types according to Cottin et al. Considering that IPF is the most widely studied and most common ILD, we separated it as a dependent category to make the comparisons more detailed. Changes have been made in the methods, results, discussion, and conclusion sections.

Reviewer: 3

Dr. Pierre-Antoine Juge, Hopital Bichat - Claude-Bernard

Comments to the Author:

In this observational study, authors retrospectively collected 6MWT from patients with ILD of various cause in order to validate the 6MWT outside IPF and to investigate the correlation between the variables of the 6MWT and the cardiopulmonary function (NYHA scale, FVC, FEV, DLCO). If the 6MWT is already used in ILD outside IPF, only literature have investigated this test in ILD outside IPF. The great number of included patients (680) having various causes of ILD makes it original. As authors stated, the main limitation of this study is the lack of investigation of an association with prognosis/mortality. Here are my comments:

Dear Dr. Pierre-Antoine Juge, thank you for reviewing our work. We appreciate your constructive suggestions, which helped us to improve the manuscript. To facilitate your reading, we marked the revisions in blue.

MAJOR COMMENTS

1. Authors have conducted this study because of the lack of data about 6WT in ILD outside IPF. They used the classification of Cottin et al in order to classify ILD causes. I think it would be useful to separate IPF from the other idiopathic ILD and analyze them separately.

Response to the reviewer: We agree with the reviewer's comment here. We have separated IPF out as a category and have re-analyzed the data. Changes have been made in methods, results, table 3, discussion, and conclusion sections.

2. Distance was measured using meters. However, it varies widely according to age and sex. Authors could transform the data to %of expected in a same sex/same age population using official references.

Response to the reviewer: Thanks for your constructive suggestion. We have calculated and applied the ratio of 6MWD and predicted 6MWD as a key variable to do the subsequent analysis. Data were re-analyzed, and related results were added. Changes have been made in the methods section, results section, table 4, and discussion section.

3. Authors give the results from heart rate and SpO₂ at 6 min and at the end of the test. What is the difference between these 3 measures? When was the measure at the end of the test assessed?

Response to the reviewer: Sorry for the unclear. We have unified the variable names throughout the manuscript, which are at the 0 min, the end of 1 min, the end of 2 min, the end of 3 min, the end of 4 min, the end of 5 min, and the end of 6 min. Changes have been made throughout the manuscript, tables, and figure 1.

4. Methods: Please specify included time and location of the included patients (multicenter? Monocenter?). The methods should specify how diagnosis was performed (expert pulmonologist? Radiologist involved?).

Response to the reviewer: Thanks. We have supplemented the recruitment time, location of the center, and how the diagnosis was established in the methods section.

5. Could the authors specify when the PFTs were performed and the NYHA scale collected regarding the 6MWT? In another way, was the PFTs performed the same day than the 6MWT? Same comment for NYHA data.

Response to the reviewer: Thanks for reminding us to supplement this information. In this study, NYHA and PFTs were assessed during hospitalization. According to the treatment routine of the ward, the hospitalized patients would complete all the assessments and tests within three days of admission. Therefore, the outcomes of NYHA, PFTs, and 6MWT were obtained within the adjacent three days. We have supplemented this in the methods section.

6. Limitations: replication in another population (or multicentric study) could have reinforce the message. Please comment in the discussion.

Response to the reviewer: We agree with the reviewers' comments. We have pointed out this in both discussion and limitations sections.

MINOR COMMENTS

1. Abstract: please rephrase the objective. It does not seem to me that it reflects well the main objective of the study (investigate the 6MWT and its variable among ILD in and outside IPF and their correlation with NYHA and PFT).

Response to the reviewer: We have revised the objective according to your suggestions. Changes have been made in the abstract and introduction sections.

2. I would have moved Table 3 to supplementary

Response to the reviewer: We got several significant differences between groups after re-analyzing the data. Therefore, we decided to remain with Table 3 in the main text.

3. Table 1 and 4. Please specify the unit used for each variable. Female (or Male) sex can be removed from the table (1 is sufficient because the other becomes implicit)

Response to the reviewer: We have added the unit of each variable in all tables and removed the female from Table 1.

4. Please change Gender to Sex

Response to the reviewer: We have changed gender to sex.

5. Please have the manuscript English rewritten. Overall, the manuscript is well written but there are small mistakes (past/present tense, missing words, hard to understand phrasing...)

Response to the reviewer: Thanks for your comment. We have improved our manuscript writing throughout with the help of a professional editing service.

VERSION 2 – REVIEW

REVIEWER	Ayed, Khadija University of Tunis El Manar, physiologie
REVIEW RETURNED	31-Mar-2022
GENERAL COMMENTS	I recommend acceptance and publication of this manuscript.